# HAFNet: Hierarchical Attentive Fusion Network for Multispectral Pedestrian Detection

**Peiran Peng [1]**, **Tingfa Xu [1,2,3]**, **Bo Huang [4]** and **Jianan Li [2,3,*]**

1    The School of Optics and Photonics, Beijing Institute of Technology, Beijing 100081, China
2    Key Laboratory of Photoelectronic Imaging Technology and System, Ministry of Education of China, Beijing 100081, China
3    Chongqing Innovation Center, Beijing Institute of Technology, Chongqing 401135, China
4    College of Optoelectronic Engineering, Chongqing University , Chongqing 400044, China
*    Correspondence: lijianan@bit.edu.cn

**Abstract:** Multispectral pedestrian detection via visible and thermal image pairs has received widespread attention in recent years. It provides a promising multi-modality solution to address the challenges of pedestrian detection in low-light environments and occlusion situations. Most existing methods directly blend the results of the two modalities or combine the visible and thermal features via a linear interpolation. However, such fusion strategies tend to extract coarser features corresponding to the positions of different modalities, which may lead to degraded detection performance. To mitigate this, this paper proposes a novel and adaptive cross-modality fusion framework, named Hierarchical Attentive Fusion Network (HAFNet), which fully exploits the multispectral attention knowledge to inspire pedestrian detection in the decision-making process. Concretely, we introduce a Hierarchical Content-dependent Attentive Fusion (HCAF) module to extract top-level features as a guide to pixel-wise blending features of two modalities to enhance the quality of the feature representation and a plug-in multi-modality feature alignment (MFA) block to fine-tune the feature alignment of two modalities. Experiments on the challenging KAIST and CVC-14 datasets demonstrate the superior performance of our method with satisfactory speed.

**Keywords:** multispectral pedestrian detection; content-dependent; feature alignment

## 1. Introduction

Pedestrian detection is a challenging computer vision task and has been widely used in urban scenes [1,2]. With the rapid development of artificial intelligence technology, pedestrian detection has become a major research focus in the field of computer vision. Applications such as autonomous driving [3] and remote surveillance [4] require accurate detection performance in challenging urban environments, where factors such as insufficient illumination and occlusion [5] pose significant challenges. Pedestrian detection using mono-spectral images as the source of information is particularly challenging. For example, in low-light or foggy conditions, pedestrians in visible images may blend into the background, while in infrared modality, thermal background noises with similar heat levels to pedestrians can lead to false detection. Figure 1 illustrates the benefits of using multispectral images over visible-only or thermal-only images in various scenarios, highlighting their superiority in challenging conditions such as low light or thermal background noise.

At present, with the tremendous advances in convolutional neural networks (CNNs), two-stream CNN-based detectors are widely used in the field of multi-modality pedestrian detection [5–9]. The classical CNN-based two-stream detector consists of three parts, which are a two-branch feature extraction module to extract the modality features, a feature fusion and augmentation module to blend and enhance the features of both modalities, and a detection module for decision-making. The system takes pairs of visible-thermal images as inputs and outputs the joint detection results for each image pair. Given that visible

and thermal images capture different characteristics of objects, leveraging these modalities can significantly improve the performance of object detection. On the one hand, visible cameras capture sophisticated visual nuances (such as color and texture information) in unobstructed and well-illuminated environments. On the other hand, thermal cameras are sensitive to temperature variations, which provides a significant advantage for detecting objects in occluded or poorly-lit environments.

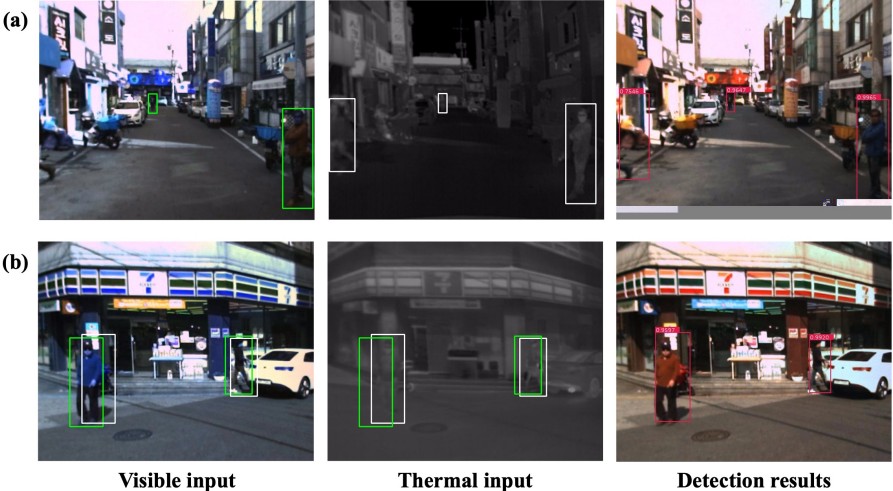

**Figure 1.** Illustration of inter-modality complementarity and dislocations. (**a**) modality complementarity, where thermal features complement the impairments of visible features. (**b**) Object dislocations, where the misalignment of the two modes causes the quality of the fused features to be impaired. Our HAFNet eliminates the misalignment of objects between visible and thermal modalities and adaptively fuses the corresponding features to boost detection.

Multi-modality feature representations are diverse at the same spatial location, but they all correspond to the same object. Therefore, fusing the visible features with the thermal features while utilizing this valuable information can potentially enhance detection performance. In well-illuminated settings, visible cameras can capture intricate visual nuances, such as color and texture information, while thermal cameras can detect objects in ill-lit or occluded environments by sensing temperature variations. To achieve this, there were two major challenges: (i) How to obtain ultra-complete feature representations of object without the interference of noise using visible and thermal essential information? Since the reliability of visible and thermal features varies under different illumination conditions, direct fusion may not only complement feature information but also fuse noise, which may lead to degradation of detection performance. Therefore, how to integrate the representations to fully exploit the inherent complementary information between the modalities and how to design an effective feature fusion mechanism to achieve maximum performance gain are still open questions to be investigated. (ii) How to eliminate the misalignment of features in visible and thermal modalities? If different modality features are fully aligned, we can directly perform pixel-wise feature fusion to enhance the feature representation. However, because the multi-modality features are often misaligned, direct fusion operations may not provide any benefit to improving detection performance and may even hinder detection. Figure 1a shows an example of modality complementarity to remove noise interference in the visible and thermal modalities (e.g., object incomplete, occlusion). Figure 1b shows an example of the dislocation of visible and thermal modalities. The noise and misalignment in both modalities may be superimposed and worsen the robustness and accuracy, leading to detection failure.

To address these issues, current approaches mainly focus on leveraging the internal complementarity between modalities to enhance object feature representation through fusion mechanisms. Previous works [8,10,11] utilized the internal information comple-

mentarity between the modalities to enhance the feature representation of the object via a fusion mechanism. The standard fusion framework first extracts the CNN features of the two modalities separately and then uses the fusion strategy to obtain the features that contain the features of both modalities. To explore the optimal stage for performing fusion, researchers from KAIST [12] created a new multispectral pedestrian dataset and evaluated the detection results of different fusion stages. MBNet [11] introduces a differential modality-aware fusion module to overcome the unbalance issue between modalities during fusion. CFT [13] first adopts Transformer for multispectral object detection, which integrates global contextual information from different modalities in the feature extraction backbone. These approaches propose solutions from different perspectives. We believe that the existing methods do not fully exploit the potential of attentional information to enhance fusion features due to the lack of implicit modeling of correlations between modalities. Furthermore, the crucial issue of pixel-level alignment of multi-modality features is often ignored, leading to the assumption that the features of both modalities are perfectly aligned by default. This can weaken the effectiveness of feature fusion as the network propagates forward.

To overcome the aforementioned problems, we present a novel and efficient framework as shown in Figure 2, named Hierarchical Attentive Fusion Network (HAFNet). The HAFNet embedded with a hierarchical content-dependent attentive fusion (HCAF) module and a multi-modality feature alignment (MFA) block to leverage multi-modality knowledge at the same location after aligning the multi-modality features and to obtain enhanced pedestrian features. Specifically, inspired by the attention mechanism, the MFA block exploits the correlation between the two modalities by first fine-tuning the thermal features to align with the visible features during feature extraction at each stage. Next, the HCAF merges the top-level feature maps of both modalities and scales them to the same resolution as the features of each stage to serve as reference features, which is done to eliminate background noise interference and increment the hierarchical features. The HCAF then utilizes the reference feature map as a guide to perform pixel-wise blending of the two modality features at each stage to enhance the quality of the feature representation in the final detection. Finally, the enhanced fused features from the framework are fed into the detection module to detect the pedestrians.

Extensive experiments conducted on several benchmarks demonstrate that HAFNet achieves high detection performance and robustness, especially in challenging scenarios with occlusion or low-light environments. To sum up, this work makes the following contributions:

- A novel Hierarchical Attentive Fusion Network (HAFNet) is proposed, enabling the progressive calibration of features from two modalities, resulting in an improved fusion representation.
- A novel module called Hierarchical Content-dependent Attentive Fusion (HCAF) is presented, which utilizes top-level features across modalities to obtain hierarchical reference features. These features are then used to guide the pixel-wise fusion of multi-modality features at each stage, resulting in improved feature alignment and integration.
- A novel Multi-modality Feature Alignment (MFA) block is proposed, which can be easily integrated into any pre-trained multi-branch backbone, enhancing the learned feature representations.
- Experimental results on the challenging datasets KAIST [14] and CVC-14 [15] demonstrate that HAFNet is competitive in terms of robustness and accuracy compared with the state-of-the-art method, while maintaining satisfactory speed.

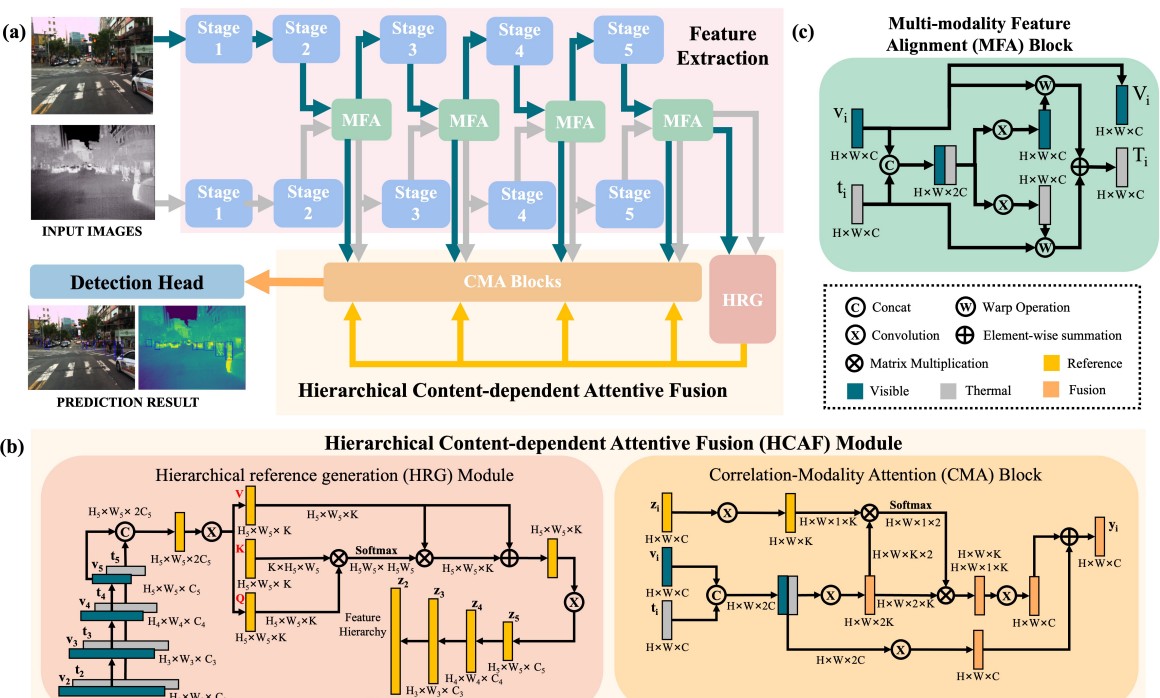

**Figure 2.** Architecture of the proposed HAFNet. The HAFNet consists of a feature extraction module plugged into a multi-modality feature alignment (MFA) block, a novel hierarchical content-dependent attentive fusion (HCAF) module, and a detection head module. (**a**) Workflow of the proposed HAFNet. (**b**) Workflow of the proposed HCAF module, which comprises a Hierarchical reference generation (HRG) module and correlation-modality attention (CMA) block. (**c**) Workflow of the MFA block.

## 2. Related Work

### 2.1. Multispectral Pedestrian Detection

The multispectral pedestrian detection field has seen significant advancements in recent years, with various fusion architectures and techniques proposed. KAIST [14] was the first to release a large-scale multispectral pedestrian detection dataset, followed by studies such as Wagner et al. [16], who found that late fusion architecture outperformed early fusion and traditional aggregated channel features (ACF) methods [10]. Halfway fusion [12] and Fusion RPN [17] also demonstrated that middle-stage fusion performs better than early or late-stage fusion. Li et al. [18] proposed a middle-level fusion network that jointly optimizes pedestrian detection and semantic segmentation tasks to improve detection performance, while Li et al. [6] and Guan et al. [19] trained a fusion network to estimate the illumination value for adaptive fusion. Zhang et al. [8] proposed CIAN, which fuses middle-level thermal and visible features under the guidance of cross-modality interactive attention. Zhou et al. [11] introduced a feature alignment module and a differential modality aware fusion (DMAF) module to select features from two-stream images according to illumination conditions. Recent studies have focused on the misalignment of multispectral pedestrian image pairs and their labels, with approaches such as AR-CNN [20] and MLPD [21] showing promising results. However, one aspect that has not been thoroughly explored in previous studies is modality-specific occlusion, which can significantly impact the performance of detection systems. However, these methods have not considered modality-specific occlusion, which can impair the final detection performance.

In this context, our proposed Hierarchical Attentive Fusion Network (HAFNet) addresses the issue of modality-specific occlusion with a hierarchically content-dependent attentive fusion (HCAF) module. HAFNet suppresses unwanted features, refines the

counterpart modality, and improves the unified representations for multispectral pedestrian detection.

### 2.2. Attention Mechanism

Attention has become a popular technique for enhancing feature representations in various applications. Channel-attention was introduced in SENet [22], while CBAM [23] proposed a network with spatial attention. Wang et al. [24] combined CNN with self-attention in a non-local network. Drawing inspiration from these methods, we treat features extracted from multi-modal source images as different transformation expressions for feature enhancement. To address the core challenges of visible-thermal pedestrian detection, which are the full utilization of differential information in both modalities and the handling of modality-specific occlusion, we propose a hierarchical cross-modality feature fusion module, the HCAF module, that leverages cross-attention knowledge. We use a hierarchical reference generation (HRG) module to tailor and design the reference feature maps. Our correlation-modality attention (CMA) module specializes in extracting common features and emphasizing complementary features under the guidance of hierarchical correlation information, thereby improving the quality of the fused representation.

### 3. Method

The proposed HAFNet is designed to process a pair of thermal-visible image patches as input. To address the issue of modality misalignment that could hinder feature fusion, we introduce a multi-modality feature alignment (MFA) module that fine-tunes the thermal features to align with visible features during feature extraction at each stage. To enhance the representation of multi-modality features and take advantage of their inherent complementary information, we propose a hierarchical content-dependent attentive fusion (HCAF) module that effectively fuses and recalibrates features of both modalities to obtain quality features. The overall architecture is illustrated in Figure 2a. Our network mainly consists of two key components, the HCAF module and the MFA module, which are described in detail below.

### 3.1. Hierarchical Content-Dependent Attentive Fusion

The proposed HCAF module is an efficient hierarchical content-dependent attentive fusion mechanism designed to fuse visible and thermal features guided by top-level information. As shown in Figure 2b, the HCAF module pixel-by-pixel selects and blends features from both modalities with the guidance of top-level features that have rich semantic information. This content-dependent fusion mechanism enhances the perception of object information and supports more accurate and stable pedestrian detection by integrating the features from both modalities.

#### 3.1.1. Formulation

Formally, let $V = \{V_1, \ldots, V_N\}$ and $T = \{T_1, \ldots, T_N\}$ denote two sets of multi-modality feature maps from multiple levels.

$$Y_i = f_{\text{cma}}(Z_i, V_i, T_i), \tag{1}$$

$$Z = f_{\text{hrg}}(V, T), \tag{2}$$

where $Z_i$ represents the reference feature map in $Z$ with the same resolution as $V_i$ and $T_i$. $f_{\text{cma}}(\cdot)$ is a feature fusion function, called correlation-modality attention (CMA), which is used to encode the reference feature map $Z$ into the multi-modality features to obtain feature maps with more accurate representation. Equation (2) represents the generation process of the reference feature map $Z$, where $f_{\text{hrg}}(\cdot)$ denotes the generation function, called hierarchical reference generation (HRG), which can produce a set of reference feature maps $Z$ by fusing the spatial features of the two modalities.

*Correlation-modality attention (CMA).* Inspired by the non-local network architecture proposed by Wang et al. [24], we introduce the Correlation-modality attention (CMA) as a feature fusion block in our proposed method. This approach enables the pixel-by-pixel fusion of features extracted from both modalities at each stage, allowing for a more robust and accurate representation of the object being detected. As shown in Figure 2b, the CMA refines the response at point $k$ in $Z_i$, $z_i^k$, as the query feature, and refines the response of $k$ points in features of multiple modalities, $\{x_i^k\}_{x=[v,t]}$, as the key feature.

$$\varphi(z_i^k, v_i^k, t_i^k) = \sum_{x \in [v,t]} \frac{e^{\theta(z_i^k)\phi_x(x_i^k)^{\mathrm{T}}}}{\sum_{\forall x} e^{\theta(z_i^k)\phi_x(x_i^k)^{\mathrm{T}}}} \phi_x(x_i^k), \tag{3}$$

$$f_{\mathrm{cma}}(Z_i, V_i, T_i) = \varphi(Z_i, V_i, T_i) + \Phi(V_i, T_i), \tag{4}$$

where $\theta(\cdot)$ is a linear embedding operation implemented by a $1 \times 1$ convolution with learnable weight matrix $W_\theta : \theta(z_i^k) = W_\theta z_i^k$. $\phi_x(\cdot)$ is also a linear embedding operation, which is implemented by $1 \times 1$ convolution with the learnable weight matrix $W_{\phi_x}$ for $x_i^k : \phi_x(x_i^k) = W_{\phi_x} x_i^k$. The fusion function $\Phi(\cdot)$ is implemented by concatenation operation, $1 \times 1$ convolution and batch normalization operation.

*Hierarchical reference generation (HRG).* To obtain reference feature maps with hierarchical information, we propose the Hierarchical Reference Generator (HRG) module. The HRG module merges the visible and thermal features at a certain stage and resizes them to the same shape as the features of each stage. The resulting feature maps are used as reference maps for the corresponding stage to guide the pixel-by-pixel fusion of the features from both modalities in the subsequent CMA block. By incorporating the HRG module, the HCAF module can effectively capture the hierarchical and semantic information of the input features and adaptively fuse them for more accurate pedestrian detection. Let $p$ represent the index of the reference feature map:

$$Z_p = g(V_p, T_p), \tag{5}$$

$$f_{\mathrm{hrg}}(V, T) = \psi(Z_p, V, T), \tag{6}$$

where $z_p$ denotes the reference feature map obtained by features at stage $p$. A transformation function $\psi(\cdot)$ obtains the reference feature maps $Z$ by resizing $Z_p$ to the same shape as each feature map in $V$ by a bilinear interpolation operation or a dilated max pooling operation. Whether to utilize reference feature maps with hierarchical information will be discussed in the ablation study.

The feature fusion function $g(\cdot)$ obtains a reference feature map that contains the object features enriched from the two modalities. We consider three instantiations of $g(\cdot)$ as follows:

*Spatial attention.* First, following the self-attention form [25], $g(\cdot)$ can be defined as an attention operation to concatenate feature map of the both modalities. In order to achieve additional modeling of local spatial context, a separable depth-wise convolutional projection layer $\varepsilon(\cdot)$ is performed on the concatenation feature map as query, $Q$, key, $K$, and value, $V'$.

$$Q, K, V' = \varepsilon(V'_i \oplus T_i), \tag{7}$$

$$g(V_p, T_p) = Softmax(\frac{QK^{\mathrm{T}}}{\sqrt{d}})V', \tag{8}$$

where $d$ denotes the dimension of the key. $\oplus$ indicates a concatenation operation of visible-thermal features in the channel dimension.

***L*₁-*norm.*** We also consider fusing features between modalities though a $l_1$-norm strategy [26]. $g(\cdot)$ can be defined as a combination of $l_1$-norm and *softmax* operation:

$$g(V_p, T_p) = \frac{\xi(V_p) \times V_p + \xi(T_p) \times T_p}{\xi(V_p) + \xi(T_p)}, \qquad (9)$$

where the $l_1$-norm operation $\xi(\cdot)$, evaluates the spatial-wise average value: $\xi(\cdot) = e^{\|(\cdot)\|_1}$.

*Concatenation.* Alternatively, one can also define *g* as a direct fusion operation.

$$g(V_p, T_p) = \tau(V_i \oplus T_i), \qquad (10)$$

where the feature fusion function $\tau(\cdot)$ is implemented by a concatenation operation and a linear embedding operation. Whether to utilize reference feature maps with hierarchical information will be discussed in the ablation study.

The multiple available instantiation forms of $g(\cdot)$ illustrate the flexibility of the generation of the reference feature maps, and the performance of each instantiation will be discussed in the ablation study.

### 3.1.2. Implementation Design

We implement HCAF as an adaptive feature fusion module that blends and reinforces features from both modalities. In this way, it can be flexibly stitched behind a two-branch backbone to extract an information-rich feature representation of the object.

Extending ResNet [27] to a visible-thermal branch backbone as an example, Figure 2b shows the architecture of an instance of the HRG module implementing $g(\cdot)$ as spatial attention. We obtain $V$ and $T$ by collecting the feature maps from the feature extraction module at stages 2 to 5, denoting $\{V_2, V_3, V_4, and V_5\}$, and $\{T_2, T_3, T_4, and T_5\}$, respectively. The stages of the reference feature map fusion are dynamically selected based on the semantic information, and we perform Stage 5 as an example to obtain the reference feature map $Z_5$. The setting of the fusion stage to obtain the reference feature map will be discussed in ablation studies. The reference feature map $Z_5$ is then resized to obtain the reference feature maps $Z$, where each feature map has the same resolution as that in $V$. Finally, the refined fusion feature maps $Y$ are fed into the detection module by fusing the feature maps in $V$ and $T$ at the same stage, under the guidance of the reference feature map $Z$.

### *3.2. Multi-Modality Feature Alignment*

To address the issue of feature mismatch in multispectral images that may affect the effectiveness of pixel-level feature fusion, we propose a novel modality alignment mechanism. This mechanism modulates thermal and visible features by generating a semantic flow that contains cross-modality properties. By aligning the features at the pixel level, the fusion process can be performed more effectively. Our proposed mechanism can be incorporated into the backbone network to align the features of both modalities before they are passed to the fusion module.

Based on the above considerations, we design an efficient plug-in multi-modality feature alignment (MFA) block, as shown in Figure 2c, that will be inserted into the feature extraction module, replacing the original thermal features with aligned ones to support accurate and stable detection.

### 3.2.1. Formulation Details

Specifically, given the intermediate convolution feature maps of two modalities $V = \{V_1, \ldots, V_N\}$ and $T = \{T_1, \ldots, T_N\}$. The aligned thermal features $T'$ can be expressed as follow:

$$T'_i = \vartheta(\zeta_V(V_i \oplus T_i), \zeta_T(V_i \oplus T_i)), \qquad (11)$$

where $\oplus$ denotes a concatenation operation of visible-thermal features in channel dimension. $\zeta(\cdot)$ represents a convolution operation with a $3 \times 3$ kernel, where different superscripts

represent different convolution kernel parameters. The warp function $\vartheta(\cdot)$ indicates the position adjustment operation of the feature [28] and $T'_i$ denotes the calibrated thermal feature maps of stage $i$.

### 3.2.2. Block Design

To enhance the alignment of features from different modalities, we introduce the MFA module as a plug-in block that can be easily integrated into any pre-trained multi-branch backbone. The MFA module, as shown in Figure 2c, is designed to align the thermal features with visible features using a content-dependent approach. We initialize the convolutional weights of the MFA module with a Gaussian distribution with a zero mean at the start of training to ensure that the module can perform identity mapping without changing the behavior of the pre-trained backbone.

### 3.2.3. Integration into Backbone CNNs

A MFA block can be flexibly plugged into arbitrary pre-trained multi-branch backbones to lift their learned feature representation. Extending ResNet [27] as an example, we obtain $V$ and $T$ by collecting the feature maps before the last residual block in stages 2 to 5, denoted $\{V_2, V_3, V_4, and V_5\}$, and $\{T_2, T_3, T_4, and T_5\}$, respectively. In Stage 4, for example, we adopt $V_4$ as the aligned target feature map to adjust $T_4$ since the visible features contain richer position information of the object. The block outputs a aligned feature map $T'_4$, upon which subsequent feed-forward computations in the backbone are conducted normally.

### 3.3. Optimization

To generate the reference feature map $Zi$, we optimize the top reference feature map $ZN$ in an end-to-end manner using a binary cross-entropy loss function called $L_{\text{bce}}$. This loss function approximates the mask map obtained from the ground truth bounding box. The ground truth mask map $\vec{Z}\text{ref}$ and the reference loss function $L_{\text{ref}}$ can be formulated as follows:

$$\vec{Z}_{\text{ref}} = \psi(\delta(y_{\text{v}}, y_{\text{t}}, Z_{\text{N}}), \tag{12}$$

$$L_{\text{ref}} = L_{\text{bce}}(\vec{Z}_{\text{ref}}, Z_{\text{N}}), \tag{13}$$

where $y_{\text{v}}$ and $y_{\text{t}}$ represent labels from visible and thermal images. The label merging function, $\delta(\cdot)$, combines the multi-modality labels corresponding to the same image without duplicating the labels to generate the final truth bounding box. An interpolate operation $\psi(\cdot)$ resizes the label and generates a mask map $\vec{Z}_{\text{ref}}$ with the same resolution as $Z_{\text{N}}$.

Our classification and localization loss terms are based on the ones used in RetinaNet [29]. Specifically, we employ sigmoid focal loss ($L_{\text{cls}}$) for classification and L1 loss ($L_{\text{reg}}$) for localization. Finally, the final loss term $L$ is the weighted sum of the three loss terms, as follows:

$$L = \lambda_1 L_{\text{ref}} + \lambda_2 L_{\text{cls}} + L_{\text{reg}}, \tag{14}$$

where $\lambda_1$ and $\lambda_2$ are weight factors to balance three loss terms, and $L_{\text{cls}}$ and $L_{\text{reg}}$ denote the loss terms for classification and localization. We set $\lambda_1$ to 0.1 and $\lambda_2$ to 1 in our experiments.

## 4. Experiments

### 4.1. Implementation Details

*Parameter setting.* Our detector extends the ResNet [27] backbone and utilizes the 5 residual blocks and batch normalization layers of its pre-trained model on ImageNet as the initial parameters for our visible-thermal branch. The remaining convolutional layers are initialized with a normal distribution, with the value of the standard deviation set to 0.01; the other parameters of our model are initialized using the Xavier approach [30]. For the reason that the vast majority of pedestrians can be represented by a vertical bounding box, we set the anchor box as $1/1$ and $1/2$ for aspect ratios, $[2^0, 2^{1/3}, 2^{2/3}]$ for fine scales and 40, 80, 160, 240 for scales levels. We train the network on 4 Nvidia GTX 1080Ti with a

batch size of 8. Furthermore, we use Stochastic Gradient Descent (SGD) as the optimizer, setting the initial learning rate as 0.0001, momentum as 0.9 and weight decay as 0.0005. As for data augmentation, the input image size is resized to $512 \times 640$ and we use the random horizontal flips, random crops, and to increase diversity.

*Evaluation metric.* We use the standard miss rate ($MR^{-2}$) as a representative score. This is the most popular metric for pedestrian detection tasks, with lower scores indicating better performance. This metric only focuses on a high-precision region rather than a low-precision region, and as such, it is more appropriate for commercial solutions. Furthermore, to better illustrate the performance of our proposed method, we also follow the standard evaluation of the false positive per image (FPPI) [10], which is sampled in the range of $[10^{-2}, 10^0]$.

### 4.2. Results on KAIST Dataset

*Data and setup.* The KAIST Multispectral Pedestrian Dataset [14] consists of 95,328 fully-overlapped visible-thermal pairs under different illumination conditions. The provided ground truth consists of 103,128 pedestrian bounding boxes in 1182 instances. Considering the faulty annotations in the training dataset, we follow the standard criteria provided by ref. [31] that a total of 25,076 frames are used for training. For evaluation, we utilize the sanitized annotations [18], which contains 2252 frames consisting of 1455 frames in day time and 797 frames in night time. This is the standard criterion for a fair comparison with recent related works. Additionally, only pedestrians taller than 50 pixels are considered.

*Main results.* The proposed HAFNet is evaluated and compared with 12 state-of-the-art methods in nine subsets of the KAIST dataset (i.e., all, day, and night for the time subsets, near, medium, and far for the scale subsets, and none, partial, and heavy for the occlusion subsets). For a fair comparison with recent works, we apply the same protocol to all methods. As illustrated in Table 1 and Figure 3, the HAFNet shows better performance compared with other competing methods.

**Table 1.** Miss rate comparison on nine subsets of the KAIST dataset [16]. Sca. and Occ. denote scale and occlusion, respectively.

| Method | Miss Rate (%) | | | MR-Scale (%) | | | MR-Occlusion (%) | | |
|---|---|---|---|---|---|---|---|---|---|
| | All | Day | Night | Near | Medium | Far | None | Partial | Heavy |
| ACF [14] | 47.32 | 42.57 | 56.17 | 28.74 | 53.67 | 88.20 | 62.94 | 81.40 | 88.08 |
| Halfway Fusion [12] | 25.75 | 24.88 | 26.59 | 8.13 | 30.34 | 75.70 | 43.13 | 65.21 | 74.36 |
| Fusion RPN + BF [17] | 18.29 | 19.57 | 16.27 | 0.04 | 30.87 | 88.86 | 47.45 | 56.10 | 72.20 |
| MSDS-RCNN [18] | 11.63 | 10.60 | 13.73 | 1.29 | 16.19 | 63.73 | 29.86 | 38.71 | 63.37 |
| IAF-RCNN [6] | 15.73 | 14.55 | 18.26 | 0.96 | 25.54 | 77.84 | 40.17 | 48.40 | 69.76 |
| IATDNN + IAMSS [19] | 14.96 | 14.67 | 15.72 | 0.04 | 28.55 | 83.42 | 45.43 | 46.25 | 64.57 |
| CIAN [8] | 14.12 | 14.77 | 11.13 | 3.71 | 19.04 | 55.82 | 30.31 | 41.57 | 62.48 |
| AR-CNN [31] | 9.34 | 9.94 | 8.38 | 0.00 | 16.08 | 69.00 | 31.40 | 38.63 | 55.73 |
| MBNet [11] | 8.13 | 8.28 | 7.86 | 0.00 | 16.07 | 55.99 | 27.74 | 35.43 | 59.14 |
| MLPD [21] | 7.58 | 7.95 | 6.95 | - | - | - | - | - | - |
| BAANet [32] | 7.92 | 8.37 | 6.98 | 0.00 | 13.72 | 51.25 | 25.15 | 34.07 | 57.92 |
| RISNet [33] | 7.89 | 7.61 | 7.08 | 0.00 | 14.01 | 52.67 | 25.23 | 34.25 | 56.14 |
| HAFNet (Ours) | 6.93 | 7.68 | 5.66 | 0.00 | 13.68 | 53.94 | 26.31 | 30.10 | 55.16 |

Moreover, the proposed framework ranks first in six of the nine subsets. For instance, in the night subset, our HAFNet (5.66%) surpasses MLPD (6.95%) by 1.29%, which demonstrates the robustness of HAFNet against noise in low-light environments at night. This also demonstrates that the framework can effectively suppress low-light noise in visible features and obtain superior quality features by fusing efficient object information in thermal features. In addition, the HAFNet also outperforms in all subsets of pedestrian occlusion. Especially in the partial occlusion scenario, the HAFNet (30.10%) surpasses BAANet (34.07%) by 3.97%, which indicates the remarkable ability of our HAFNet to address oc-

clusion in multispectral pedestrian detection. These results evidence that the proposed HAFNet is able to generate high-quality fused features by preserving rich mode-specific information and excluding noise.

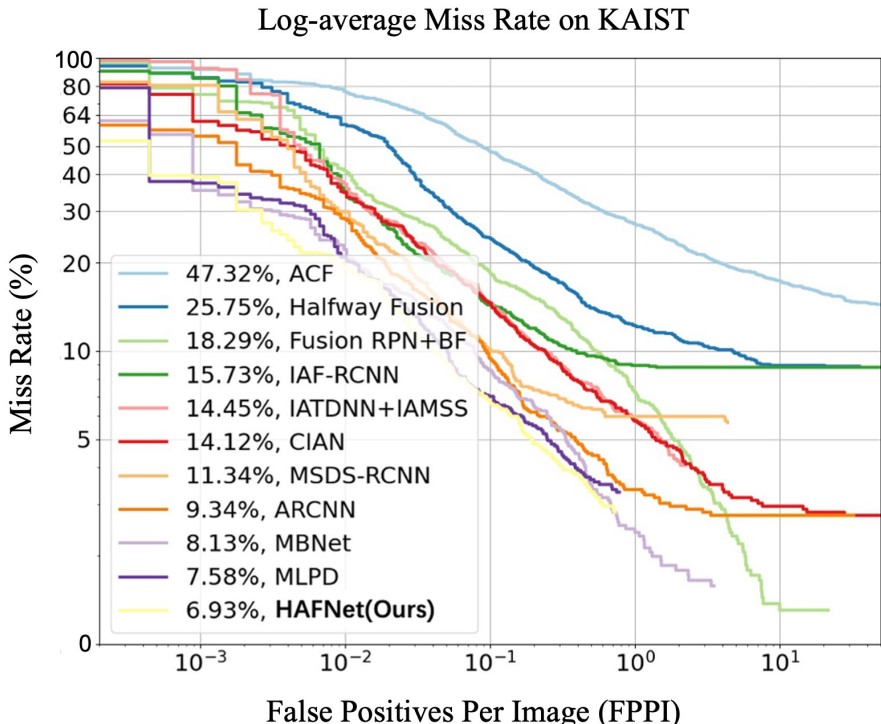

**Figure 3.** Comparison of the detection results with the state-of-the-art methods on the KAIST dataset under the sanitized subset.

*Qualitative results.* The detecting examples of our HAFNet with other detectors are shown in Figure 4. It can be seen that HAFNet performs outstandingly during both the day and night. In cases of poor illumination or partial occlusion pedestrians, the HAFNet still locates the pedestrians accurately. Figure 5 demonstrates the superiority of our proposed method over the baseline. In the first two lines of the figure, it is evident that the baseline method can only detect relatively large targets in a well-lit environment with no occlusion, while our method can detect relatively small objects in a low-light environment. The attention maps in the third and fourth lines of the figure show that our method accurately captures the features of the pedestrian even in the presence of partial occlusion. Notably, even in the case of heavy occlusion, as shown in the sixth line of the figure, our method can extract pedestrian features and accurately detect targets by leveraging the internal connection between the two modalities.

*Speed.* The speed comparison of HAFNet and other state-of-the-art methods is shown in Table 2. Compared with other detectors at the same level of miss rate, the speed of HAFNet still performs satisfactorily. The speed versus miss rate of all methods is illustrated in Figure 6, which shows the proposed methods perform remarkably well in balancing miss rate and speed. Compared with previous methods, we achieve an advantage in accuracy and show satisfying results in terms of speed.

**Table 2.** Speed comparison between HAFNet and state-of-the-art methods. It should be noted that ACF and Fusion RPN+BF did not provide information on the hardware platform used in the original paper.

| Methods | MR-All (%) | Platform | Speed (s) |
| --- | --- | --- | --- |
| ACF [14] | 47.32 | MATLAB | 2.730 |
| Fusion RPN + BF [17] | 18.29 | MATLAB | 0.800 |
| CIAN [8] | 14.12 | GTX 1080Ti | 0.070 |
| AR-CNN [31] | 9.34 | GTX 1080Ti | 0.120 |
| MBNet [11] | 8.13 | GTX 1080Ti | 0.070 |
| MLPD [21] | 7.58 | GTX 1080Ti | 0.012 |
| BAANet [32] | 7.92 | GTX 1080Ti | 0.070 |
| HAFNet(Ours) | 6.93 | GTX 1080Ti | 0.017 |

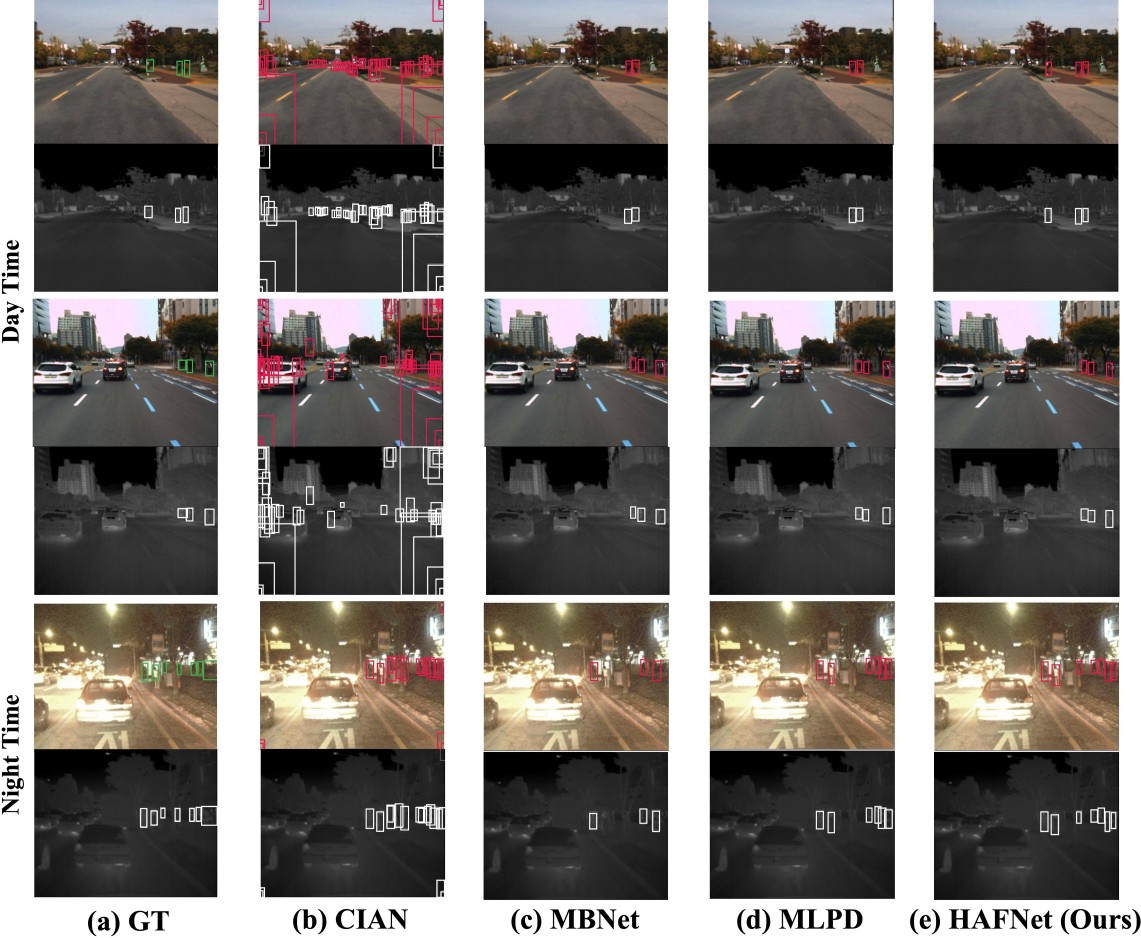

**(a) GT**　　　**(b) CIAN**　　　**(c) MBNet**　　　**(d) MLPD**　　**(e) HAFNet (Ours)**

**Figure 4.** Qualitative evaluation of HAFNet with three top ranked methods, i.e., CIAN [8], MBNet [11] and MLPD [21]. The green rectangles denote the ground truth, while the red ones denote the detection results. (**a**) Ground truth. (**b**) Results of the CIAN [8]. (**c**) Results of the MBNet [11]. (**d**) Results of the MLPD [21]. (**e**) Results of our HAFNet.

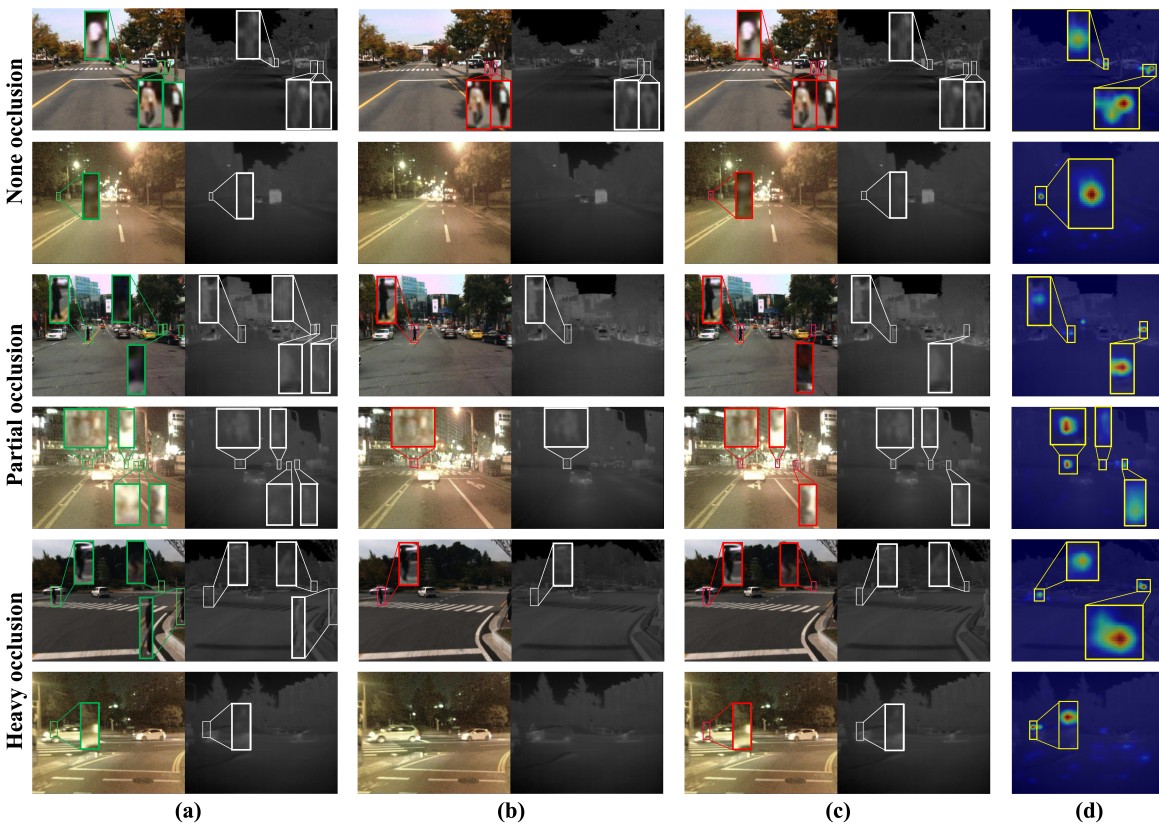

**Figure 5.** Visualizing the performance comparison of baseline and proposed HAFNet detection models under three distinct occlusion scenarios: none, partial, and heavy. (**a**) Ground truth of the visible-thermal image pairs. (**b**) Results of the baseline. (**c**) Results of our HAFNet. (**d**) Attention maps of our HAFNet.

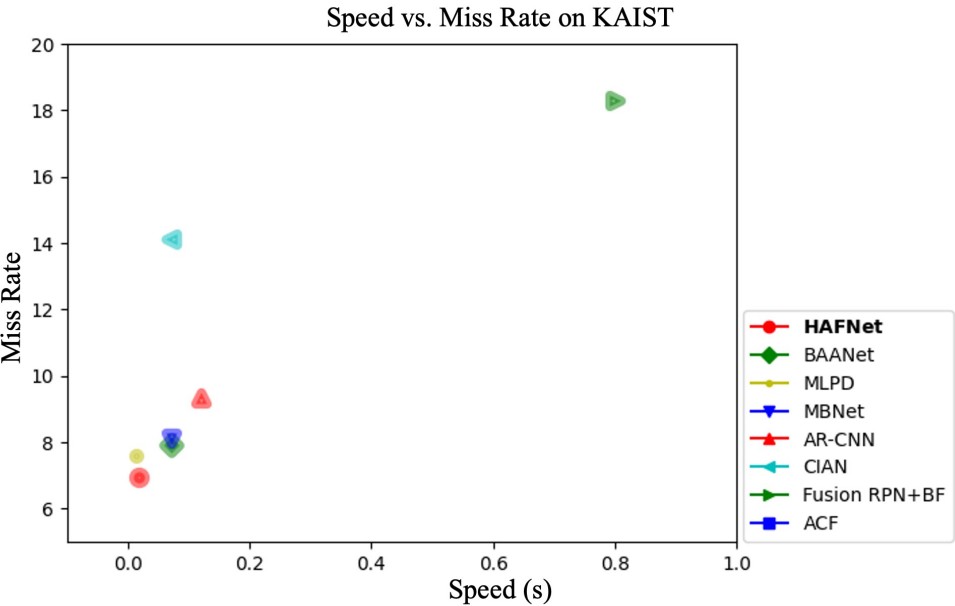

**Figure 6.** Miss rate versus the speed of our proposed HAFNet. HAFNet achieves excellent performance in terms of speed and miss rate.

### 4.3. Results on CVC-14 Dataset

*Data and setup.* The CVC-14 [15] dataset is a multispectral pedestrian dataset taken with a stereo camera configuration. The dataset contains visible (gray scale) and thermal paired images, of which 7, 085 and 1, 433 frames are for training and test sets, and provides individual annotations in each modality. However, the authors of the dataset release the cropped image pairs without the non-overlapped areas. Therefore, we treated this dataset as a fully-overlapped (paired) dataset, but it still suffers from the pixel-level misalignment problem. Moreover, there are some other issues, such as inaccurate ground truth boxes, incorrect extrinsic parameters, and unsynchronized capture systems. Nevertheless, this dataset has been used by many in works [8,11,21,34,35] because it is one of the few practical datasets captured in a stereo setup.

*Main results.* Similar to previous methods [11], we perform fine-tuning using the KAIST pretrained model during the training phase on the CVC-14 dataset. To evaluate the robustness under paired conditions, we adopt this dataset and compare our results with other methods [8,11,21,34,35]. In order to ensure a fair comparison, we adhere to the protocol established in MACF [34], which has also been adopted by other studies.

By fusing information from both modalities, our method can leverage the unique strengths of each modality to mitigate noise interference and obtain more accurate object information. This is particularly important in real-world scenarios where single modality information may be incomplete or inaccurate due to environmental factors such as occlusions or low lighting conditions. As presented in Table 3, the results indicate that the modalities-fusion approach generally outperforms the unimodal approach. Compared with the other multi-modality methods listed in Table 3, our method achieves the highest detection accuracy in all-weather and daytime scenarios. By selectively screening and fusing the unique features of both modalities, our method is able to obtain more accurate object information. Although our nighttime detection performance is slightly lower than that of MBNet [11], this still demonstrates that our network effectively leverages object features and mitigates noise interference by fusing information from both modalities. Additionally, our method demonstrates robustness in the face of a significant number of feature misalignments in the CVC-14 dataset. This highlights the ability of our approach to overcome real-world challenges that may significantly affect detection performance.

**Table 3.** Evaluation results comparison on the CVC-14 dataset. We use the reimplementation of ACF, Faster R-CNN, MACF, and Halfway Fusion in literature [34].

| Modalities Input | Methods | Miss Rate (%) | | |
|---|---|---|---|---|
| | | Day | Night | All |
| Visible only | SVM [15] | 37.6 | 76.9 | - |
| | DPM [15] | 25.2 | 76.4 | - |
| | Random Forest [15] | 26.6 | 81.2 | - |
| | ACF [34] | 65.0 | 83.2 | 71.3 |
| | Faster R-CNN [34] | 43.2 | 71.4 | 51.9 |
| Visible + Thermal | MACF [34] | 61.3 | 48.2 | 60.1 |
| | Choi et al. [35] | 49.3 | 43.8 | 47.3 |
| | Halfway Fusion [34] | 38.1 | 34.4 | 37.0 |
| | Park et al. [34] | 31.8 | 30.8 | 31.4 |
| | AR-CNN [8] | 24.7 | 18.1 | 22.1 |
| | MLPD [21] | 24.18 | 17.97 | 21.33 |
| | MBNet [11] | 24.7 | 13.5 | 21.1 |
| | HAFNet (Ours) | 23.9 | 14.3 | 20.7 |

### 4.4. Ablation Study

To verify the effectiveness of our proposed HAFNet, comprehensive ablation studies that are conducted on the KAIST multispectral pedestrian dataset are performed in this

subsection. We first validate the effectiveness of the components in our framework, then we analyze the structure and parameter settings of each component.

### 4.4.1. Ablations on Network Components

We analyze the effectiveness of different network components, i.e., Hierarchical Content-dependent Attentive Fusion (HCAF) and Multi-modality Feature Alignment (MFA) in Table 4. We design a baseline model consisting of an extending ResNet50 [27] as a two-branch backbone and a simple fusion module [31], which achieves 12.68% miss rate. For a fair comparison, as with the baseline, we performed the same training parameters and pre-processing on the same hardware platform. First, we replace the simple fusion module with our HCAF module with a 3.24% performance improvement, which demonstrates the effect of our content-dependent fusion in the feature merging and augmentation steps. Furthermore, the miss rate improved by 2.35% after only adopting MFA blocks plugged into the two-branch backbone (10.33%), which indicates that the feature alignment during feature extraction contributes to the quality of subsequent modality fusion features. The entire HAFNet, consisting of HCAF and MAF, achieves more significant performance, with a boost of 5.75% to the miss rate at 6.93% over baseline results. The results demonstrate that both proposed modules benefit multispectral pedestrian detection through the correlation between both modalities, and their combination achieves even greater results.

**Table 4.** Ablations on HAFNet. HCAF: Hierarchical content-dependent attentive fusion. MFA: Multi-modality feature alignment. CMA: Correlation-modality attention. HRG: Hierarchical reference generation.

| HCAF | | MFA | Miss Rate (%) | | |
|:---:|:---:|:---:|:---:|:---:|:---:|
| CMA | HRG | | All | Day | Night |
| | | | 12.68 | 13.70 | 10.76 |
| ✓ | ✓ | | 8.31 | 9.03 | 6.67 |
| | | ✓ | 10.33 | 12.18 | 7.53 |
| ✓ | | ✓ | 7.93 | 9.87 | 8.36 |
| | ✓ | ✓ | 8.70 | 9.53 | 7.64 |
| ✓ | ✓ | ✓ | 6.93 | 7.68 | 5.66 |

### 4.4.2. Discussion on HCAF

The structure of our HCAF consists of a feature fusion function and hierarchical reference generation, to further verify the effect of the parts of the structure, i.e., correlation-modality attention (CMA) and hierarchical reference generation (HRG).

*Impact of CMA.* The CMA is performed as a feature enhancement block guided by a reference feature map, and its effect is shown in Table 4. The network without CMA blocks shows a 1.77% reduction in the miss rate, indicating that pixel-by-pixel fusion of features from both modalities guided by the reference feature map can focus more on the features of the object, which helps in obtaining quality features.

To further verify the performance of the CMA block, we conducted experiments with different numbers of CMA blocks, as shown in Table 5. We compare our CAM on the concatenation of both modalities, with the self-attention (SA) module without pixel-wise fusion guided by reference features. We realize that more CAM blocks yield better detection performance. It is shown that guidance by higher-level semantic features to integrate the features of both modalities pixel by pixel is essential to construct an efficient detector.

**Table 5.** Ablation on the quantity of correlation-modality attention (CMA) blocks. The HAFNet utilizes feature maps from the feature extraction module in stages 2 to 5.

| | HCAF | Miss Rate (%) | | | Speed (s) |
|---|---|---|---|---|---|
| | | **All** | **Day** | **Night** | |
| HAFNet | SA(4) + HRG | 8.70 | 9.53 | 7.64 | 0.045 |
| | SA(3) + CMA(1) + HRG | 8.28 | 9.35 | 6.01 | 0.035 |
| | SA(2) + CMA(2) + HRG | 7.73 | 8.35 | 6.50 | 0.027 |
| | SA(1) + CMA(3) + HRG | 7.28 | 7.92 | 5.91 | 0.021 |
| | CMA(4) + HRG | 6.93 | 7.68 | 5.66 | 0.017 |
| HAFNet * | CMA(5) + HRG | 7.11 | 7.68 | 6.40 | 0.053 |

* The CMA in HAFNet utilizes feature maps from the feature extraction module in stages 1 to 5.

*Impact of HRG.* The HRG module, consisting of a feature fusion module and a reference feature map generation module, has a significant impact on the detection performance, as shown in Table 4. The network without the HRG module shows a 1% increase in the miss rate, demonstrating the importance of utilizing the reference feature map to improve the quality of fused features.

To explore the optimal setting of the HRG module, three instantiations of the feature fusion function were tested to obtain a reference feature map at Stage 5, as shown in Table 6. The results show that the highest detection accuracy was achieved with spatial attention as the reference feature generation method, with a 1.34% improvement over the concatenation method. Further experiments were conducted to explore the generation phase of the reference feature maps, as shown in Table 7. The results indicate that the higher the stage at which the reference feature maps are generated, the better the accuracy achieved. This demonstrates that more semantic information can effectively guide feature fusion on both modalities. In particular, the miss rate reaches 7.04% when the reference feature maps are obtained at Stage 4, which is comparable to the performance of the proposed HAFNet at Stage 5. This suggests that the HRG module can be effectively used to improve feature fusion in multispectral pedestrian detection.

**Table 6.** Ablation on the hierarchical reference generation (HRG) module.

| | HRG | Miss Rate (%) | | | Speed (s) |
|---|---|---|---|---|---|
| | | **All** | **Day** | **Night** | |
| HAFNet | concatenation | 8.27 | 8.88 | 7.07 | 0.011 |
| | concatenation * | 8.36 | 9.88 | 5.68 | 0.012 |
| | $L_1 -$ norm | 7.70 | 8.80 | 5.69 | 0.021 |
| | $L_1 -$ norm * | 8.07 | 9.10 | 5.82 | 0.035 |
| | spatial attention | 6.93 | 7.68 | 5.66 | 0.017 |
| | spatial attention * | 7.38 | 7.74 | 6.45 | 0.028 |

* Reference feature maps with hierarchical information are not utilized.

**Table 7.** Ablation on the Stage to generate the reference feature map.

| | HRG | Reference Stage | Miss Rate (%) | | | Speed (s) |
|---|---|---|---|---|---|---|
| | | | **All** | **Day** | **Night** | |
| HAFNet | spatial attention | 2 | 7.71 | 8.64 | 5.98 | 0.026 |
| | | 3 | 7.53 | 8.21 | 6.08 | 0.021 |
| | | 4 | 7.04 | 7.94 | 5.82 | 0.019 |
| | | 5 | 6.93 | 7.68 | 5.66 | 0.017 |

*Impact of hierarchical information.* To further investigate the impact of hierarchical information on detection performance, we designed experiments as shown in Table 6. From the results, it is clear that the HRG module, which utilizes the hierarchical information, can better guide the feature fusion in each stage (an improved 0.45 % miss rate on the spatial attention method).

### 4.4.3. Discussion on Stages of MFA

Observing Table 8, we can see that the HAFNet obtains the highest performance when 4 MFA blocks are plugged into the feature extraction module. The detection performance is highest when the MFA block is only plugged in the last four stages, which demonstrates the soundness of our design.

**Table 8.** Ablation on the multi-modality feature alignment (MFA) block. The MFA block is plugged into the feature extraction module of HAFNet from stage 2 to stage 5.

|  | Volume of MFA | Miss Rate (%) | | |
| --- | --- | --- | --- | --- |
|  |  | **All** | **Day** | **Night** |
| HAFNet | 1 | 8.17 | 8.84 | 6.97 |
|  | 2 | 7.69 | 8.62 | 5.86 |
|  | 3 | 7.16 | 7.56 | 6.23 |
|  | 4 | 6.93 | 7.68 | 5.66 |
| HAFNet * | 5 | 7.01 | 7.71 | 5.54 |

* The MFA block is plugged into the feature extraction module of HAFNet from stage 1 to stage 5.

### 4.4.4. Discussion on Modalities Input

We present an experiment to investigate the complementarity of multi-modal features in pedestrian detection. Specifically, we conduct detection using only visible light input, only thermal imaging input, or a combination of both modalities. Our results, shown in Table 9, demonstrate the superiority of the multi-modal approach over single-modal approaches. The fusion of visible and thermal images achieves significantly better performance in pedestrian detection, highlighting the complementary nature of the two modalities in this task. These findings provide important insights for the development of effective pedestrian detection systems in real-world scenarios.

**Table 9.** Results of ablation experiments on single-modality and multi-modality inputs for pedestrian detection.

|  | Modalities Input | Miss Rate (%) | | |
| --- | --- | --- | --- | --- |
|  |  | **All** | **Day** | **Night** |
| HAFNet | visible only | 26.49 | 17.76 | 44.61 |
|  | thermal only | 19.72 | 23.30 | 12.53 |
|  | visible + thermal | 6.93 | 7.68 | 5.66 |

## 5. Conclusions

In this paper, we tackle the challenges of integrating visible and thermal modalities with distinct characteristics and overcoming modality-specific occlusion in visible and thermal imaging. To address these issues, we propose a novel and adaptive framework called Hierarchical Attentive Fusion Network (HAFNet). By incorporating the Hierarchical content-dependent attentive fusion (HCAF) module and multi-modality feature alignment (MFA) blocks, HAFNet enables the network to suppress modality-specific occlusion features and learn the adaptive fusion of multi-modality features. Our experiments demonstrate that HAFNet achieves outstanding performance in both accuracy and speed. We believe that our work can make valuable contributions to the advancement of multispectral applications.

## 6. Discussion

The proposed HAFNet method for multispectral pedestrian detection effectively addressing issues such as modal noise and pixel misalignment while improving the quality of features for modal fusion. The HCAF module and MFA blocks effectively inhibit modality-specific occlusion features and learn the adaptive fusion of multi-modality features, resulting in outstanding accuracy and speed. Moving forward, we believe that our method can be extended to other multi-modality tasks beyond image and thermal sensing. For instance, it can be applied to speech-to-text or image-to-text tasks. We envision that our method can be modified to align and fuse the features of different modalities to enhance the performance of multimodal models. This can be a promising direction for future research, especially given the increasing interest in multi-modality applications in various fields.

**Author Contributions:** Conceptualization, J.L. and P.P.; methodology, J.L. and P.P.; software, P.P.; validation, P.P. and B.H.; formal analysis, P.P.; resources, T.X.; data curation, P.P.; writing—original draft preparation, P.P.; writing—review and editing, J.L., B.H. and P.P.; visualization, P.P.; supervision, J.L.; project administration, T.X.; funding acquisition, T.X. All authors have read and agreed to the published version of the manuscript.

**Funding:** This research received no external funding.

**Data Availability Statement:** Data is unavailable due to privacy restrictions.

**Conflicts of Interest:** The authors declare no conflict of interest.

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
