# Peer review of "HAFNet: Hierarchical Attentive Fusion Network for Multispectral Pedestrian Detection"

_remotesensing, doi:10.3390/rs15082041_

Round 1

Reviewer 1 Report

This paper proposed a multispectral pedestrian detection method named "Hierarchical Attentive Fusion Network (HAFNet)". The Hierarchical Content-dependent Attentive Fusion (HCAF) and Multi-modality feature alignment (MFA) are used to detect pedestrians. The proposed method is significant for the application of multispectral pedestrian detection, while there are more details should be further explained.

1.       Figure 2 (a), the "HCMA" module is not explained in the text. Maybe it’s the "CMA" module mistakenly wrote as "HCMA".

2.       Please further check the position of Correlation-modality attention (CMA) and Hierarchical reference generation (HRG) described in the network, as shown in Figure 2(a).

3.       The models of ACF and Fusion RPN+BF mentioned in Table 2 were tested on MATLAB platform, while the other models were tested on 1080TI GPU?

4.       The experimental results of the CVC-14 dataset should be further explained.

Reviewer 2 Report

In this work, a pedestrian detection approach based on Hierarchical Attentive Fusion Network is proposed for visible and thermal image pairs. The HAFNet comprises MFA module, HCAF module and detection head, which are used to adjust thermal features, guide the pixel-wise multi-modal feature integration, and detect pedestrians, respectively. Experiments shows the robustness and accuracy superiorities. Generally, the paper is well written. I have several issues that hope the authors will take into consideration.

1) Paragraph “Spatial attention”, “defined as a attention” => “an”

2) Paragraph “Block Design”, convolutional weight parameters are initialized to zero? I cannot understand the mechanism. Please explain this point.

3) In the loss function L, what is the difference between Lref and Lcls/Lreg’? Aren’t they derived from the ground truth map? I think sub-section 3.4 should be re-organized in a clearer style.

4) The bounding box on the thermal image of Fig. 4 is not clear enough. Authors are suggested to choose a high-contrast color.

5) Fig. 4 and Fig. 5 cannot reveal the occlusion cases. It is necessary to show the none, partial, and heavy occlusion cases in a more obvious fashion.

6) The results of CVC-14 dataset are short. It is recommended to present more results and analyses.

7) This work fuse visible and thermal images for pedestrian detection. In my opinion, it is necessary to show the detection results using single image modality.

Reviewer 3 Report

This paper introduces a new technique for multispectral pedestrian detection. The experiment results are interesting. The manuscript is well organized and generally clear. However, the manuscript has to be improved before publication. My comments are as follows:

1. Generally, the introduction section represents the significance, the background, the gap, and the methodology of the research. It is recommended to rewrite the introduction section considering the information mentioned above. In addition, there is a need to introduce the research gap in detail and to show how the authors deal with it.

2. Please avoid to write the results in the introduction section.

3. Please avoid the repetition of “we introduce”.

4. I suggest to the authors to detail more the related works section in order to give a clear view to the readers.

5. Please edit figure 2 to make more understandable for the readers.

6 Although it is not necessary for the publication of the paper in my opinion, I may recommend to add a discussion section.

7. Please rewrite the method section to make it easier to understand for the readers.

Round 2

Reviewer 2 Report

The authors have properly addressed my comments. I think this version is deserved to be published. The authors might also present graph results in 4.4.4 if possible. Thank you.